# Botulinum Toxin A Injection for the Treatment of Intractable Dry Eye Disease

**DOI:** 10.3390/medicina57030247

**Published:** 2021-03-08

**Authors:** Eun Woo Choi, Dong Ju Yeom, Sun Young Jang

**Affiliations:** 1Department of Ophthalmology, Soonchunhyang University Bucheon Hospital, Soonchunhyang University College of Medicine, 170, Jomaru-ro, Bucheon 14584, Korea; chenw333@naver.com; 2Gangnam Smile Eye Clinic, 6F, Tongyeong Building, 405, Gangnam-daero, Seocho-gu, Seoul 06615, Korea; rest00@naver.com

**Keywords:** botulinum toxin A, dry eye disease, periocular injection

## Abstract

*Background and Objectives:* To evaluate the clinical efficacy of periocular botulinum toxin A (BTA) injection in patients with intractable dry eye disease (DED). *Materials and Methods*: Medical records of patients with intractable DED who underwent periocular BTA injection from December 2019 to March 2020 were reviewed retrospectively. Patients were injected with 2.5 units of BTA in the medial part of the lower eyelids. The clinical data collected included age, sex, ocular surface disease index (OSDI) score, tear film break up time (TBUT), Schirmer test results, tear osmolarity (I-PEN), and tear meniscus height (TMH) measured by anterior segment optical coherence tomography. All subjective and objective data were collected before treatment and at 1 month after treatment. *Results:* Twenty-eight consecutive patients were eligible for chart review and analysis. Significant improvements in OSDI, tear osmolarity, and TMH were observed at 1 month after periorbital BTA injection. At the baseline and 1-month follow-up examinations, OSDI scores were 62.22 ± 21.30 and 47.98 ± 17.23, respectively (*p* < 0.001). TMH increased significantly after treatment (82.25 ± 40.50 at baseline vs. 138.02 ± 66.62 1-month after treatment; *p* < 0.001). Tear osmolarity using I-PEN showed a significant decrease after treatment (320.82 ± 24.66 at baseline vs. 302.75 ± 22.33 at 1 month after treatment; *p* < 0.001). No significant differences were found in TBUT or Schirmer test results before and after BTA injection. *Conclusions:* BTA injection into the medial part of the eyelid improves dry eye symptoms, the amount of tear retention, and tear osmolarity. Based on the objective parameters of the tear condition, this study supports the idea of BTA use as a potential treatment option for patients with intractable DED.

## 1. Introduction

Dry eye disease (DED) has been defined recently as a “multifactorial disease of the ocular surface characterized by a loss of homeostasis of the tear film, and is accompanied by ocular symptoms in which tear film instability and hyperosmolarity, ocular surface inflammation and damage, and neurosensory abnormalities play etiological roles, according to the International Dry Eye Workshop II [1]. Severe dry eye can cause disabling pain and fluctuating vision, significantly affecting the vision-related quality of life and limiting daily activities [1,2]. To date, numerous efforts have been made to overcome DED.

Botulinum toxin A (BTA) is a neurotoxin that blocks the release of acetylcholine at the neuromuscular junctions of cholinergic nerves [3]. Therapeutic administration of the toxin reduces muscle contractions. Most humans blink about 12 times per minute. The blink reflex distributes tears over the ocular surface and is very important as it protects the eye against foreign objects contaminated with microbes [4]. In 1997, Spiera et al. [5] first reported increased tear levels on the Schirmer test after treating essential blepharospasm with botulinum toxin in patients with Sjögren’s syndrome. In 2000, Sahlin et al. [6] found decreased mean blink-time output in patients with dry eyes after botulinum toxin injections, indicating that the injection of botulinum toxin could reduce the efficacy of lacrimal drainage by paralyzing the orbicularis oculi muscles. Injecting botulinum toxin into the pretarsal orbicularis muscle has been reported to be effective in superior limbic keratoconjunctivitis, one of the characteristics of DED [7]. More recently, prospective randomized double-blinded interventional studies to determine the effects of BTA injection on DED have been published [8,9].

Many recent studies have shown that botulinum toxin injection is effective for dry eyes; however, most of these studies were based on patients’ DED subjective symptoms or objective test results obtained from simple devices. Furthermore, Schirmer test results on these patients after periorbital BTA injection have been inconsistent, showing lower and higher values following treatment [8,9,10].

In this study, we analyzed the effect of BTA injection on the improvement of severe dry eye symptoms based on the subjective symptoms of patients and the numerical effects of various imaging devices and instruments that have been newly introduced to assess the tear condition.

## 2. Materials and Methods

This is a retrospective observational study that included 56 eyes of 28 DED patients who were treated between December 2019 and March 2020. The protocol for the study were approved by the Institutional Review Board of Soonchunhyang University Bucheon Hospital, Bucheon, Korea (IRB No.2021-01-007, 28 January 2021). The study was designed and implemented in full accordance with the principles of the Declaration of Helsinki.

All patients suffered from intractable DED. Their dry eye symptoms did not improve, despite treatment with steroid eyedrops and artificial tears, for at least 2 months. The persistent dry eye symptoms included ocular pain, dryness, and positive fluorescein corneal staining. The subjects were offered periocular BTA injections to treat their dry eye; only patients who provided informed permission received injections.

All patients received the injection with 2.5 units of BTA (Innotox; Medytox Inc., Seoul, Korea) in the medial part of the eyelid (Figure 1). The botulinum toxin concentration was 2.5 IU/0.1 mL. An injection was given near the punctum and was directed along the medial canthal tendon. All injections were performed by one operator (SYJ).

Patient data were recorded pre-injection and 1 month thereafter. One eye per patient was randomly selected for analysis. Dry eye was evaluated using both subjective and objective methods. Eye discomfort was evaluated based on the results of an ocular surface disease index (OSDI) questionnaire used to quantify dry eye symptoms. Subjects responded to the symptoms of dry eye that they had experienced during the previous week. The OSDI questions consisted of three subscales: ocular symptoms, vision-related functions, and environmental triggers, with each answer scored on a four-point scale from 0 (no problem) to 4 (significant problem). The score was calculated using the OSDI formula: OSDI = sum of scores × 25/number of questions answered [11].

The objective tear film evaluation was based on the tear film break up time (TBUT), the Schirmer tear secretion test and I-PEN (I-PEN Osmolarity System; I-MED Pharma Inc., Dollard-de-Ormeaux, QC, Canada) results, and anterior-segment optical coherent tomography (AS-OCT; Carl Zeiss Meditec Cirrus HD OCT) measurements of the precorneal tear film height [12]. The TBUT was defined as the time required for dry spots of fluorescein staining on the corneal surface to appear after blinking; the mean of three measurements was recorded. Schirmer’s test measured tear production in millimeters after placing a Schirmer strip in the lower conjunctival sac between the external third of the sac and the lateral area for 5 min, without instillation of a local anesthetic.

All comparisons were performed with a Student t-test. Data are expressed as the mean ± standard deviation. Statistical analyses were performed using SPSS statistical software for Windows (version 26.0; SPSS Inc., Chicago, IL, USA). A *p*-value of less than 0.05 was considered to indicate statistical significance.

## 3. Results

Twenty-eight consecutive patients were eligible for chart review and analysis. A total of 56 eyes of 28 subjects completed the 1-month visit. The mean age was 56.64 years (range, 36–71 years), and there were 24 women (86%). There were 14 subjects with Sjögren’s syndrome (50%). The mean OSDI was 62.22 ± 21.30, and all subjects had a pathological OSDI score of >13. All patients were using hyaluronic-based tear substitutes at the time of the analysis; 18 (64%) were applying cyclosporine eyedrops twice daily both before and after analysis, while 6 (21%) were applying fluorometholone-based steroid eye drops four times daily to control their dry eye symptoms.

The ocular surface parameters before and at 1 month after BTA injection are shown in Table 1. Tear meniscus height (TMH) increased significantly from 82.25 ± 40.50 to 138.02 ± 66.62 (*p* < 0.001). A representative AS-OCT image of one of the subjects is shown in Figure 2.

Our results showed significant improvements in the OSDI score and tear osmolarity. At baseline and at the 1-month follow-up, OSDI scores were 62.22 ± 21.3, and 47.98 ± 17.23, respectively (*p* < 0.001). Tear osmolarity using I-PEN was reduced significantly after treatment (320.82 ± 24.66 before vs. 302.75 ± 22.33 after treatment, *p* < 0.001). No significant differences were found in the TBUT (3.07 ± 0.97 vs. 3.30 ± 1.06, *p* = 0.229), and Schirmer test results (6.11 ± 2.51 vs. 7.18 ± 4.13, *p* = 0.100) before and after BTA injection (Table 1).

No subjects complained of any specific side effects after injection. Twenty patients (71.4%) requested a second BTA injection.

## 4. Discussion

In the present study, we investigated the efficacy of periorbital BTA injection to improve DED refractory to conventional treatments. The results showed significant improvements in OSDI, tear osmolarity, and TMH at 1 month after treatment. Thus, we suggest that periorbital BTA injection improved dry-eye related symptoms by increasing the retention tear amount. Specifically, BTA injection into the medial part of the eyelid improved dry eye symptoms, the amount of tear retention, and tear osmolarity.

Botulinum toxin injection has been suggested as a treatment option for DED, as it reduces lacrimal drainage [13]. Paralysis of the orbicularis oculi muscle affects the canaliculi, decreasing the function of pumping during blinking. This decrease in function increases the tear retention time [13]. These findings were supported by those of Park et al. [14] regarding changes in tear production, distribution, and drainage after BTA injection in 23 patients with essential blepharospasm; in the study, TMH was measured by OCT and dacryoscintigraphy using technetium pertechnetate-99m to assess lacrimal drainage quantitatively.

As mentioned above, previous studies have shown inconsistent results of the Schirmer test [5,8,9,10,15,16,17]. The studies in which the injection sites were limited to the medial side of lower and upper eyelids have shown increased Schirmer test values [8,9]. In our study, the Schirmer test results did not show a statistically significant difference. However, other parameters related to tear amount, such as TMH, revealed a statistical difference after BTA injection. Although the Schirmer test is a well-standardized test, there is a lack of high-level evidence data on repeatability, sensitivity, and specificity for the Schirmer test [18,19].

Although some authors have reported that the Schirmer test with topical anesthesia or nasal stimulation might be more objective and reliable in DED detection, there is a lack of high-level evidence data on repeatability, sensitivity, and specificity for these approaches. Sahlin et al. [6] investigated the effect of botulinum toxin injection on tear drainage in patients with DED; their study showed that limited dose injection produces better results with fewer complications, such as eyelid retraction. Paralysis of the orbicularis oculi muscle caused by botulinum toxin blocks the function of the lacrimal pump, reducing the release of tear fluid into the lacrimal duct, resulting in more tear fluid remaining on the ocular surface [9]. In addition, the reduction of the blinking force induced by botulinum toxin can improve symptoms by reducing blinking and micro-trauma of the corneal epithelium, inflammation of the ocular surface, and loss of membrane-related mucins in the corneal epithelium [20]. However, since botulinum toxin injected near the lacrimal gland can spread to the glands and affect tear production, it is important to consider the injection location and dosage. In the present study, all patients received an injection of 2.5 units of BTA in the medial parts of the eyelid (Figure 1), which is a commonly used treatment protocol for blepharospasm.

The BTA dose and the injection site were selected based on previous studies [6,8]. In one of these studies [6], complications such as purulent conjunctivitis were reported after injection. However, no specific complications were observed in the current study. This may be related to differences in the operator’s technique, and/or the small sample size of our study.

In this study, patient data were recorded pre-injection and 1 month thereafter. However, the effects of BTA persist for 4 months. Thus, a longer follow-up might have provided more useful clinical information. Patients require repeat BTA injections to maintain the therapeutic effects. In the present study, 20 patients (71.4%) requested a second BTA injection. Bukhari [13] compared the efficacy of BTA and punctal plugs in 60 patients with DED. In their study, all patients who received BTA injections were satisfied with the treatment results; however, only 72.3% of the patients who received punctal plugs were satisfied with their outcomes. In another report that compared BTA injection and punctal plugs for the treatment of DED [15], the authors reported that the BTA injection group had a lower complication rate than the punctal plug group (25% vs. 60%). In the present study, two patients suffered from punctal plug-related canaliculitis before BTA injection. Punctal plug insertion is simple, effective, safe, and non-invasive; however, complications still exist with this treatment [21]. Thus, patients with a history of punctal plug complications would be likely to benefit from periorbital BTA injection.

## 5. Conclusions

BTA injection into the medial part of the eyelid improves dry eye symptoms, the amount of tear retention, and tear osmolarity. Based on the objective parameters of the tear condition, this study supports the use of BTA as a potential treatment option for patients with intractable DED.

## Figures and Tables

**Figure 1 medicina-57-00247-f001:**
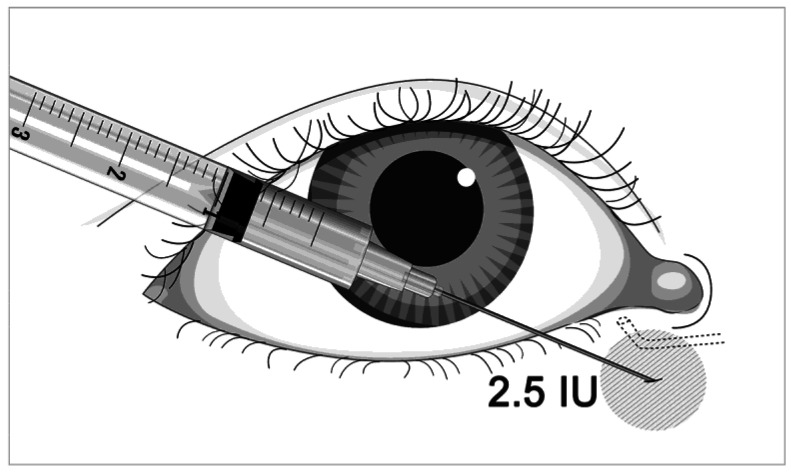
Schematic diagram illustrating periorbital botulinum toxin A injection.

**Figure 2 medicina-57-00247-f002:**
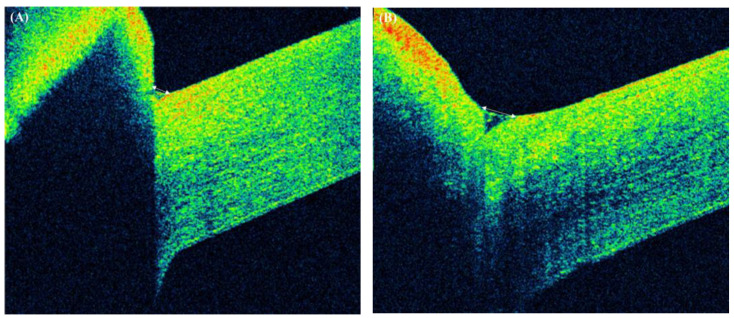
Representative images of anterior-segment optical coherence tomography of a 54-year-old female. Note the increase in tear meniscus height after botulinum toxin A injection: (**A**) before treatment and (**B**) after treatment.

**Table 1 medicina-57-00247-t001:** Comparison of OSDI, tear osmolarity, Schirmer test, TMH, and TBUT before and 1 month after BTA injection.

	Before	1 Month after BTA Injection	*p* Value
OSDI score	62.22 ± 21.30	47.98 ± 17.23	<0.001
Tear osmolarity (mOsm/L) *	320.82 ± 24.66	302.75 ± 22.33	<0.001
Schirmer test (mm)	6.11 ± 2.51	7.18 ± 4.13	0.100
TBUT (sec)	3.07 ± 0.97	3.30 ± 1.06	0.229
TMH (mm)	82.25 ± 40.50	138.02 ± 66.62	<0.001

OSDI: ocular surface disease index, TBUT: tear film break-up time, TMH: tear meniscus height, BTA: botulinum toxin A. * Using I-PEN.

## Data Availability

The data collected and analyzed in the study will be available from the corresponding author on reasonable request.

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
