# Peer review of "Botulinum Toxin A Injection for the Treatment of Intractable Dry Eye Disease"

_medicina, 2021, doi:10.3390/medicina57030247_

Round 1

Reviewer 1 Report

This study analyzed the effect of BTA injection on the improvement of severe DED based on the subjective symptoms of patients and the numerical effects of imaging devices and instruments that have been recently introduced to test the tear condition.

The aim of this study is very interesting and promising but there are some aspects to improve and deepen:

  1. The introduction section is not very well treated. It is missing of literature; in my opinion, you could improve this section with eye diseases narrative. A general review could help you: Current Evidence on the Ocular Surface Microbiota and Related Diseases. Petrillo F, et al. Microorganisms. 2020 Jul 13;8(7):1033.
  2.  I think that a retrospective study of only 3 months is a little bit poor. You should extend the analysis towards a longer time.

Author Response

Reviewer Comments:
Reviewer #1:

This study analyzed the effefct of BTA injection on the improvement of severe DED based on the subjective symptoms of patients and the numerical effects of imaging devices and instruments that have been recently introduced to test the tear condition. The aim of this study is very interesting and promising but there are some aspects to improve and deepen:

  1. The introduction section is not very well treated. It is missing of literature; in my opinion, you could improve this section with eye diseases narrative. A general review could help you: Current Evidence on the Ocular Surface Microbiota and Related Diseases. Petrillo F, et al. Microorganisms. 2020 Jul 13;8(7):1033.

Response: We appreciate the positive comments of reviewer #1, and the suggestion. We now cite the abovementioned reference in the Introduction, and have revised the text as follows:

Page 2, Introduction: Botulinum toxin A (BTA) is a neurotoxin that blocks the release of acetylcholine at the neuromuscular junctions of cholinergic nerves [3]. Therapeutic administration of the toxin reduces muscle contractions. Most humans blink about 12 times per minute. The blink reflex distributes tears over the ocular surface, and is very important as it protects the eye against foreign objects contaminated with microbes [1].

Additional citation:

[1] Petrillo, F.; Pignataro, D.; Lavano, M.A.; Santella, B.; Folliero, V.; Zannella, C.; Astarita, C.; Gagliano, C.; Franci, G.; Avitabile, T.; et al. Current evidence on the ocular surface microbiota and related diseases. Microorganisms 2020, 8.

  1. I think that a retrospective study of only 3 months is a little bit poor. You should extend the analysis towards a longer time.

Response: Given the limitations inherent to retrospective studies, we cannot expand the analysis to cover a longer time period. However, we fully agree with the reviewer. Thus, we have noted in the Discussion that a longer follow-up study is necessary; the short follow-up time is indeed a limitation of the present study.

Page 5, Discussion: In this study, patient data were recorded pre-injection and 1 month thereafter. However, the effects of BTA persist for 4 months. Thus, a longer follow-up might have provided more useful clinical information. Patients require repeat BTA injections to maintain the therapeutic effects. In the present study, 20 patients (71.4%) requested a second BTA injection.

We sincerely thank the reviewer for taking the time and effort to review our article. We hope that the revised manuscript is deemed suitable for publication in Medicina Journal.

Yours faithfully,

Sun Young Jang, MD.

Reviewer 2 Report

Although the subject is interest, there are some points that has to be solved before publishing:

Major concerns:

Is there any wash out prior to treatment? Sentence line 104 is not clear. Is it at the beginning or during the study?

Patients could use any artificial tear? Which type? How many used them?  Were steroid eye drops also used? There were any other inclusion or exclusion criteria?

Only one eye per patient could be used to not disturb results. Normally they are selected randomly prior before start the trial, but should be done anyway. It is not stated if now is using one or both eyes.

Paragraph 4 and 5 of discussion must be connected and grouped into only one because it is almost the same in the two final sentences.

Line 139 Why there is no evidence of using dacryoscintigraphy as mentioned here.

Minor concerns:

Line 129. “We believe” is not a scientific expression.

Line 146 Schimer “test” no text.

Join sentences 67 and 68.

Author Response

Reviewer #2:
Although the subject is interest, there are some points that has to be solved before publishing:

Response: We agree that several points required clarification. We have addressed these concerns in a point-by-point manner, and have edited the text accordingly. We appreciate the very careful review of Reviewer #2.

Major concerns:

  1. Is there any wash out prior to treatment? Sentence line 104 is not clear. Is it at the beginning or during the study?

Response: This was a retrospective observational study, not a randomized controlled study. Thus, there was no washout prior to treatment. We have clarified this as follows:

Page 2, Materials and Methods: This was a retrospective observational study. All patients suffered from intractable DED. Their dry eye symptoms did not improve, despite treatment with steroid eyedrops and artificial tears, for at least 2 months. The persistent dry eye symptoms included ocular pain, dryness, and positive fluorescein corneal staining. The subjects were offered periocular BTA injections to treat their dry eye; only patients who provided informed permission received injections.

  1. Patients could use any artificial tear? Which type? How many used them? Were steroid eye drops also used? There were any other inclusion or exclusion criteria?

Response: We have added clinical data on the artificial tears and prior treatments.

Page 3, Results: The mean OSDI was 62.22 ± 21.30, and all subjects had pathological OSDI scores >13. All patients were using hyaluronic-based tear substitutes at the time of analysis; 18 (64%) were applying cyclosporine eyedrops twice daily both before and after analysis, while 6 (21%) were applying fluorometholone-based steroid eyedrops four times daily to control their dry eye symptoms.

  1. Only one eye per patient could be used to not disturb results. Normally they are selected randomly prior before start the trial, but should be done anyway. It is not stated if now is using one or both eyes.

Response: We have clarified this point as follows:

Page 3, Materials and Methods: Patient data were recorded pre-injection and 1 month thereafter. One eye per patient was randomly selected for analysis.

  1. Paragraph 4 and 5 of discussion must be connected and grouped into only one because it is almost the same in the two final sentences.

Response: We have merged paragraphs 4 and 5.

  1. Line 139 Why there is no evidence of using dacryoscintigraphy as mentioned here.

Response: We thank the reviewer. In line 139, we state that Park et al. [13] explored the changes in tear production, distribution, and drainage after BTA injection into patients with blepharospasms using both OCT and dacryoscintigraphy. However, we do not routinely perform ”dacryoscintigraphy” when evaluating dry eye patients. Again, this was a retrospective observational study. We lack dacryocystographic data.

Minor concerns:

Line 129. “We believe” is not a scientific expression.

Line 146 Schimer “test” no text.

Join sentences 67 and 68.

Response: We have corrected all of the cited errors.

Line 129: Thus, we believe we suggest that periorbital BTA injection improved dry-eye related symptoms by increasing tear retention.

We sincerely thank the reviewer for taking the time and effort to review our article. We hope that the revised the manuscript is acceptable for publication in Medicina Journal.

Yours faithfully,

Sun Young Jang, MD

Round 2

Reviewer 1 Report

Thanks for the revised version.